# FAM50A as a novel prognostic marker modulates the proliferation of colorectal cancer cells via CylinA2/CDK2 pathway

Longhai Li[ORCID][1]*, Zhaoshuai Ji[2]◉, Guangyun Li[3], Hao Gu[1]*, Yan Sun[4]*

**1** School of Basic Medical Sciences, Anhui Medical University, Hefei, Anhui, China, **2** Department of Pharmacy, Beijing Tsinghua Changgung Hospital, School of Clinical Medicine, Tsinghua University, Beijing, China, **3** Department of Gastrointestinal Surgery, Bozhou Hospital of Anhui Medical University, Bozhou, Anhui, China, **4** Department of Internal Medicine Oncology, Jiangsu Cancer Hospital, Nanjing, Jiangsu, China

◉ These authors contributed equally to this work.
* sunyan@188.com (YS); guhao@ahmu.edu.cn (HG); pe_lilonghai@126.com (LL).

## Abstract

### Objective

Colorectal cancer (CRC) is the third most prevalent malignant tumor type and the second leading cause of cancer-related death. Sequence similarity family 50 member A (FAM50A) plays a vital role in numerous disease processes, including tumor progression. This study aimed to evaluate the prognostic significance of FAM50A in CRC and to explore its role in CRC cell proliferation.

### Methods

TCGA and GTEX databases and immunohistochemical staining (IHC) was used to study the expression of FAM50A in CRC tissues. Patient survival data were used to assess the prognostic significance of FAM50A in CRC using Kaplan–Meier analysis and Cox regression analysis. The Cell Counting Kit-8 (CCK-8), 5-ethynyl-2′-deoxyuridine (EdU), and colony-formation assays were employed to assess the impact of FAM50A on tumor cell proliferation. Flow cytometry was used to detect the changes of cell cycle. The cell cycle and cycle-related proteins were measured via western blotting (WB) to explore the potential mechanisms involving in cancer progresses.

### Results

The results of IHC revealed a notable upregulation of *FAM50A* expression levels in CRC tissue compared with adjacent normal tissue. Moreover, *FAM50A* expression was positively correlated with N and TNM stages in 145 patients with CRC. Cox regression analysis and construction of a nomogram revealed that high FAM50A expression was a prognostic indicator for poor overall survival in patients with CRC. Knockdown of FAM50A decreased cell proliferation ability, the proportion of EdU positive cells, and the number of CRC cell colonies, whereas overexpressing FAM50A promoted proliferative phenotypes. Knocking

**Data availability statement:** The data underlying the results presented in the study are available in Supporting Information files.

**Funding:** The Bozhou Key R&D projects (No. bzzc2021020).

**Competing interests:** The authors have declared that no competing interests exist.

**Abbreviations:** 95% CI, 95% confident interval; CRC, colorectal cancer; CCK-8, Cell Counting Kit-8; EdU, 5-ethynyl-2′-deoxyuridine; FAM50A, sequence similarity family 50 member A; HR, hazard ratio; IHC, immunohistochemical; M, distant metastasis; N, nodal involvement; OS, overall survival; qRT-PCR, quantitative reverse transcription-polymerase chain reaction; r, correlation coefficient T, depth of tumor invasion; TNM, tumor-node-metastasis; WB, western blotting.

down FAM50A induced a significant increase in the number of cells in the S phase. Meanwhile, CyclinA2 and CDK2 were significantly reduced after FAM50A knocking down.

## Conclusion

FAM50A may be a novel prognostic marker for CRC, and may participate in regulating tumor progression by targeting the CyclinA2/CDK2 signal pathway.

## 1. Introduction

Colorectal cancer (CRC) ranks as the third most common malignant tumor type globally, and the second leading cause of cancer-related death [1]. The incidence and mortality rates of CRC have stabilized and declined in some Western countries because of nationwide screening programs and the widespread use of colonoscopy [2]. However, the incidence of CRC remains high, resulting in a significant financial burden to society [3]. Owing to continued economic progress and changes in dietary practices in developing countries, the number of new CRC cases is expected to increase to approximately 2.4 million globally by 2035 [4]. Despite recent progress in medical technology and the advent of multidisciplinary and comprehensive treatments for CRC, the high mortality rate associated with this disease has not improved significantly [5]. The 5-year overall survival (OS) rate is 90% in patients diagnosed with early-stage CRC, 70% in those with locally advanced CRC, and 15% in patients with metastatic CRC [6]. The majority of patients with CRC are diagnosed with an advanced stage of disease. While treatments such as chemotherapy, molecular targeted therapy, and immunotherapy have increased survival rates, some patients develop initial insensitivity or resistance to these drugs over time, leading to primary and secondary drug resistance [7–8]. Thus, the development of drug resistance remains a significant clinical challenge in the treatment of CRC [8]. The primary cause of drug resistance is typically abnormal metabolism, transport, or targeting of anti-tumor drugs [9]. Cell death pathways, cytokine pathways, oncogenic signals, compensatory feedback loop signaling pathways, and the tumor immune microenvironment can contribute to various drug resistance mechanisms [10]. Together with tumor recurrence and metastasis, drug resistance is therefore a major factor in the mortality rates related to CRC [11]. A thorough understanding of the mechanisms underlying the occurrence and progression of CRC is essential for the discovery of novel diagnostic biomarkers and treatment approaches. This will enhance our understanding of CRC and facilitate the development of more efficient clinical diagnosis and treatment methods.

Tumor progression is characterized by gradual cellular changes over time [12]. Tumor cells evolve into subtypes that have clear proliferation advantages. *In vivo*, they progressively lose the normal controls for cell proliferation and subsequently transition to a new phase of proliferation [13]. As cell proliferation and DNA mutations increase simultaneously, the cells escape their normal regulatory mechanisms. The morphology and metabolism of tumor cells are also altered. In a review published in 2011, Hanahan and Weinberg further elaborated the defining features of tumors. Cancer cells possess the ability to evade apoptosis, overcome growth suppression, induce angiogenesis, and sustain proliferative signaling, while exhibiting immortalization and unlimited replication potential [14]. Many scientists have since focused their research on tumor cell proliferation, resulting in a deeper understanding of this process [15,16].

The family of sequence similarity (FAM) refers to a group of genes that share similar sequences and are involved in various diseases [17]. Owing to the widespread expression of FAM20C in various cancer types, it has been linked to tumorigenesis in humans [18].

Furthermore, deletion of FAM13A following hypoxia was reported to decrease the proliferation and metastatic capabilities of tumor cells, suggesting that it may be a promising target for cancer therapy [19]. Sequence similarity family 50 member A (*FAM50A*) and *FAM50B* are two distinct members of the sequence similarity 50 gene family. FAM50A is known as X chromosome-associated protein 5 (Xap5) [20]. This protein has a nuclear localization sequence, and can act as a DNA-binding protein or transcription factor. FAM50A is cytogenetically positioned on human chromosome Xq28 and comprises five exons that encode 339 amino acids with a molecular weight of 40 kDa [21–22]. It may serve as a splicing factor in the processing of RNA precursors, and may also be associated with the development of X-chromosome-related intellectual disability [23]. Although a study has mentioned its expression situation in CRC [22], the function of FAM50A in CRC is not well understood, and there is limited research on its expression and role in this tumor type.

The aim of this study was, therefore, to investigate FAM50A expression during the onset and progression of CRC. First, data from The Cancer Genome Atlas (TCGA) public database was extracted and analyzed to study the potential tumorigenic effects of FAM50A in CRC. Immunohistochemical (IHC) staining was used to quantify the expression levels of FAM50A protein in CRC samples. Additionally, the prognostic significance of FAM50A in CRC was evaluated by analyzing patient survival. Then, given the importance of cell proliferation in tumor growth, several experimental methods (Cell Counting Kit-8 [CCK-8], 5-ethynyl-2′-deoxyuridine [EdU], and colony-formation assays) were used to investigate the impact of FAM50A on cell proliferation. Finally, the changes in the cell cycle following FAM50A knockdown were detected by flow cytometry and the changes of cell cycle and cycle-related proteins were determined via WB to explore the mechanisms involving in cancer progresses.

## 2. Materials and methods

### 2.1. Extraction and analysis of data from public databases

Data from public databases, TCGA and GTEx, were used to compare the mRNA expression profile of FAM50A between pan-cancer tissues and corresponding normal tissues [24]. Subsequently, RNA-seq data and corresponding clinical information from unpaired and paired samples of CRC in TCGA database were systematically collected and processed based on the research interests of our research [25]. The expression level of FAM50A was statistically analyzed following $\log_2$ transformation. The analysis and mapping of FAM50A expression in tumors was then conducted using "limma" and other R (v4.2.2) packages, with statistical significance set at $P < 0.05$ [26].

### 2.2. Immunohistochemical analysis

Building upon a previous study [27], IHC analysis was utilized to confirm the differential expression levels of FAM50A in CRC samples. Patients between September 2020 and January 2024 were collected and incorporated into the study. The dates when data were accessed for research purposes were from 01/10/2020 to 31/01/2024. Following surgical resection, the tumor diameter was measured, signs of bleeding and necrosis were noted, and relevant tissue was collected for further analysis. The samples were fixed in 10% formalin, embedded, sectioned, stained with hematoxylin and eosin (HE), and observed under a light microscope. IHC analysis was carried out on suitable samples using a rabbit anti-FAM50A antibody (Bioss, Woburn, MA, USA; bs-8208R, 1:200) using a previously described procedure [28]. The stained sections were photographed under a microscope at either 40× or 200× magnification. The expression level of FAM50A was quantified based on the intensity of staining and the proportion of positive cancer cells, as described in our previous study [29]. This study was approved

by the Ethics Committee of Bozhou Hospital, Anhui Medical University (No. 202023). Samples were collected from newly diagnosed patients with CRC who had not received any anti-tumor treatment. Written informed consent was obtained from patients prior to specimen collection.

## 2.3. Clinical correlation and prognostic analysis

The Wilcoxon test was utilized to investigate correlations between FAM50A expression levels and various clinicopathological characteristics, including T stage, N stage, M stage, clinicopathological stage, and age. Patients with CRC were stratified into high- and low-expression FAM50A groups according to the FAM50A expression level cut-off value, as determined by receiver operating characteristic (ROC) analysis. Data collection and analysis was carried out by two dedicated individuals. Following data entry and review, Kaplan–Meier analysis was employed to examine patient survival. Cox regression analysis (univariate and multivariate) was utilized to investigate the relationship between the FAM50A expression level and patient survival. A nomogram was utilized to examine the ability of FAM50A levels to determine CRC prognosis [30].

## 2.4. Cell culture and construction of cell lines with stable FAM50A knockdown and overexpression plasmids

CRC cell lines (SW480, HCT-8, HCT-116, and RKO) and human normal intestinal epithelial cell lines, FHC and HEK293T, were obtained from the American Type Culture Collection. All cells were cultured in Dulbecco's modified Eagle medium with a high glucose concentration and 10% fetal bovine serum at 37°C and 5% $CO_2$ [31]. The lentiviral vector plasmids pLKO.1-Scramble and pLKO.1-shFAM50A were generated from the lentiviral vector plasmid pLKO.1-Puro; moreover, pCDH-puro-control and pCDH-puro-FAM50A plasmids were designated sequentially [32]. The target sequences for FAM50A knockdown overexpression system were shown in Table 1. HEK293T cells were used for virus packaging and to collect viruses [33].

## 2.5. Quantitative reverse transcription polymerase chain reaction (qRT-PCR)

Total RNA was extracted from cells using a commercially available RNA extraction kit (MN-MS-RNA-250; Bioscience, Shanghai, China). The qRT-PCR assay was performed using a Prime Script RT and SYBR Premix Ex Taq kit (Vazyme, Nanjing, China) according to the

**Table 1. The primers related to FAM50A.**

| Primer name | | Sequence (5′-3′) |
| --- | --- | --- |
| sh-FAM50A-1 | Forward | CCGGCGAGATCCTTCGGAAAGACTTGGATC-CAAGTCTTTCCGAAGGATCTCGTTTTTG |
| | Reverse | AATTCAAAAACGAGATCCTTCGGAAAGACTTG-GATCCAAGTCTTTCCGAAGGATCTCG |
| sh-FAM50A-2 | Forward | CCGGGAGCTGGTACGAGAAGAACAAG-GATCCTTGTTCTTCTCGTACCAGCTCTTTTTG |
| | Reverse | AATTCAAAAAGAGCTGGTACGAGAAGAA-CAAGGATCCTTGTTCTTCTCGTACCAGCTC |
| FAM50A-overexpression | Forward | GCGAATTCATGGCTCAATACAAGGGCGC |
| | Reverse | GCGGATCCTCAGCGGATCGTGTACTTGTCC |
| FAM50A (qRT-PCR) | Forward | TCACTGAGCAACCGCACATC |
| | Reverse | GGCACCCAGCACAATGAAGA |

manufacturer's instructions. Gene expression levels were normalized to that of β-actin. The qRT-PCR results were analyzed to obtain the Ct value of gene amplification products, and the $2^{-\Delta\Delta Ct}$ method was used for further analysis. The specific primers used for qRT-PCR were shown in Table 1.

## 2.6. Western blotting

Cells were lysed using radioimmunoprecipitation assay buffer (P0013B; Beyotime Biotechnology, Shanghai, China) containing proteinase and phosphatase inhibitors. The protein concentration of the samples was measured using a bicinchoninic acid (BCA) kit (P0010; Beyotime Biotechnology, Shanghai, China) according to the manufacturer's protocol. Equal amounts of protein sample were separated using 10% sodium dodecyl sulfate-polyacrylamide gel electrophoresis and then transferred to a nitrocellulose membrane. Subsequently, the membrane was blocked with 5% skim milk powder for 1 h at room temperature (20-25°C) and incubated with the following primary antibodies overnight at 4°C: anti-FAM50A (Proteintech, Rosemont, IL, USA; 19849-1-AP, 1:1,000), CDK1 (Proteintech, 67575-1-Ig, 1:1000), CDK2 (Proteintech, 60312-1-Ig, 1:1000), CDK4 (Proteintech, 66950-1-Ig, 1:1000), CDK6 (Proteintech, 66278-1-Ig, 1:1000), Cyclin A2 (Proteintech, 66391-1-Ig, 1:1000), Cyclin B1 (Proteintech, 67686-1-Ig, 1:1000), Cyclin C (Proteintech, 67415-1-Ig, 1:1000), Cyclin D (Proteintech, 60186-1-Ig, 1:1000) and anti-β-tubulin (Proteintech, 10068-1-AP, 1:5,000). The membrane was then incubated with the appropriate horseradish-peroxidase-conjugated secondary antibody (Beyotime Biotechnology, A0208 and A0216, 1:5,000) for 30 min at room temperature. Finally, the bands were visualized using an enhanced chemiluminescence system and quantified using Image J software (National Institutes of Health, Bethesda, MD, USA).

## 2.7. CCK-8 assay

Cells were seeded (1,000 cells per well) in a 96-well plate, with three replicate wells. Following inoculation, they were incubated overnight in a controlled environment. Subsequently, 100 μL of CCK-8 solution was added to the cells and they were incubated for 1 h at five different time points (after seeding 0 [24h], 1 [24h], 2 [72h], 3 [96h] and 4 [120h] days). The optical density at 450 nm was then determined using a microplate reader (Thermo Fisher Scientific, Waltham, MA, USA) [34].

## 2.8. EdU assay

Cells were cultured in six-well plates and incubated with a final concentration of 10 μM EdU for 2 h. They were then immobilized prior to Click Additive Solution buffer, DNA staining, and image acquisition as described previously [35].

## 2.9. Colony-formation assay

For each experimental group, 1,000 cells per well were seeded into six-well plates and cultured for 10 days with careful monitoring and regular changes of the growth medium. The resulting colonies were then stained and photographed as described previously [36].

## 2.10. Flow cytometry assay

The cultured cells in 6-well plates were digested with trypsin, washed 3 times with Phosphate Buffered Saline (PBS), and then fixed with cold 70% alcohol for 30 minutes. After fixation, the cells were centrifuged again, and washed twice times with PBS. Then, 400 μL of PI staining (Beyotime Biotechnology, C1052) with moderate RNase A was added to the cube and mixed.

Subsequently, the mixture was incubated at 4 °C in the dark for 30 minutes. Finally, the cell cycles were detected and analysed by CytoFlex Flow Cytometer (Beckman Coulter, USA).

### 2.11. Statistical analysis

Descriptive and statistical analyses were conducted using Statistical Package for the Social Sciences (SPSS) version 22.0 (IBM; Armonk, New York, USA) or R software (v4.2.2). The Wilcoxon rank-sum test was utilized to assess the differential expression levels of FAM50A between the two groups. Correlations between FAM50A expression levels and clinico-pathological characteristics were determined using Spearman's rank correlation coefficient (Spearman's r). Kaplan–Meier analysis was used to examine patient survival, and groups were compared utilizing the log-rank test. One-way analysis of variance was used to compare multiple groups, and the Student's t test was used for pairwise comparisons.

## 3. Results

### 3.1. FAM50A was highly expressed in CRC

We first extracted FAM50A expression data for 33 tumor types from TCGA database. Analysis revealed that FAM50A was significantly overexpressed in various tumors, including bladder urothelial carcinoma, colon cancer, rectal cancer, and other common tumor types ($P < 0.05$, Figs 1A and 1B). Since FAM50A was highly expressed in both colon and rectal cancer, and given that CRC is our long-term research focus, this gene was selected for further detailed investigation. Subsequent analysis revealed a notable upregulation of FAM50A expression levels in CRC tumor tissue than in the adjacent normal tissue. Fig 1C depicts the results for unpaired CRC samples, while Fig 1D depicts the results for paired CRC samples. Following the observation of high FAM50A expression levels in CRC from a public database, we confirmed its expression in CRC tissue using IHC staining. A total of 145 patients with CRC were studied, comprising 67 women and 78 men. The baseline characteristics of this CRC cohort are presented in Table 2. Consistent with results of the analysis of data from TCGA database, FAM50A expression levels were higher in tumor tissue than in the adjacent normal tissue ($P < 0.05$, Figs 1E and 1F). Based on the above results, we conducted a series of in-depth studies of FAM50A.

### 3.2. FAM50A expression was related to clinical pathological features of CRC

Subsequently, we investigated correlations between FAM50A expression levels and tumor clinical characteristics. Using ROC curve analysis of scores in normal and cancer tissues (Figs 2A and 2B), 3.5 was determined as the cut-off value for distinguishing patients with CRC revealing a high (95 patients, 65.5%) or low (50 patients, 34.5%) level of FAM50A expression. Positive correlations were observed between FAM50A expression levels and several clinical characteristics, including N stage (N0 vs. N1–N2; $P < 0.001$; Fig 2J), M stage (M0 vs. M1; $P < 0.05$; Fig 2K), and TNM stage (I–II vs. III–IV; $P < 0.001$; Fig 2L). No statistically significant correlations were observed between other features (all $P > 0.05$; Figs 2C–2I). Further results are presented in Table 2. These findings suggest that FAM50A plays a crucial role in driving tumor progression and they support the hypothesis that FAM50A may act as a proto-oncogene in CRC.

### 3.3. The prognostic value of FAM50A in CRC

Patient prognosis is an important factor in clinical practice. Kaplan–Meier survival analysis was utilized to investigate the association between FAM50A expression level and prognosis

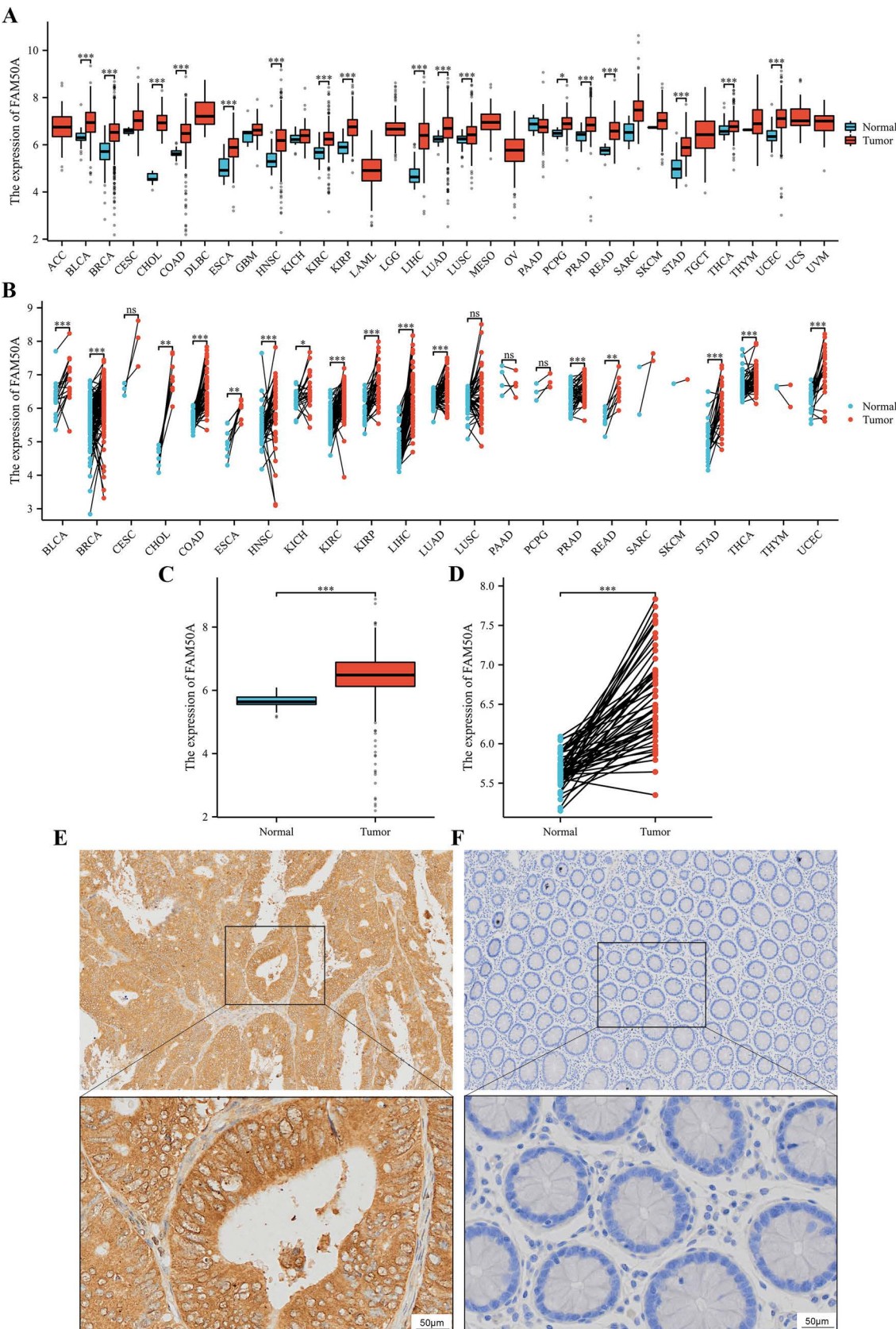

**Fig 1. High expression of FAM50A in CRC.** (A) Panoramic analysis of FAM50A gene expression in tumors and normal tissues (unpaired) from the TCGA database. (B) Panoramic analysis of FAM50A gene expression in tumors and corresponding normal

tissues (paired). (C) Expression of FAM50A in tumor tissues and normal tissues (unpaired). (D) Expression of FAM50A in paired tumor and normal tissues. (E) IHC showed high expression of FAM50A in CRC tumor cells (upper panel: 40X; lower panel: 200X). (F) Normal cells showed only weak staining for FAM50A (upper panel: 40X; lower panel: 200X). * $P \leq 0.05$, ** $P \leq 0.01$, *** $P \leq 0.001$, ns: not significant.

**Table 2. Patient baseline data involving correlation between the clinical characteristics and pathological staging in CRC patients.**

| Characteristics | Case (n = 145) | FAM50A expression | | P-value |
|---|---|---|---|---|
| | | Low | High | |
| Total | 145 | 50 | 95 | |
| Gender | | | | 0.971 |
| Male | 78 | 27 | 51 | |
| Female | 67 | 23 | 44 | |
| Age | | | | 0.302 |
| ≤60 | 81 | 25 | 56 | |
| >60 | 64 | 25 | 39 | |
| Tumor size | | | | 0.626 |
| ≤4 cm | 62 | 20 | 42 | |
| >4 cm | 83 | 30 | 53 | |
| Tumor location | | | | 0.587 |
| Colon | 68 | 25 | 43 | |
| Rectum | 77 | 25 | 52 | |
| Cancer site | | | | 0.475 |
| Left | 115 | 38 | 77 | |
| Right | 30 | 12 | 18 | |
| Differentiation | | | | 0.277 |
| Well | 78 | 30 | 48 | |
| Moderate & Poor | 67 | 20 | 47 | |
| Depth of tumor invasion | | | | 0.239 |
| T1-T2 | 46 | 19 | 27 | |
| T3-T4 | 99 | 31 | 68 | |
| Lymph node metastasis | | | | < 0.001 |
| N0 | 71 | 35 | 36 | |
| N1-N2 | 74 | 15 | 59 | |
| Distant metastasis | | | | 0.538 |
| M0 | 124 | 44 | 80 | |
| M1 | 21 | 6 | 15 | |
| TNM stage | | | | 0.004 |
| I-II | 75 | 34 | 41 | |
| III-IV | 70 | 16 | 54 | |

in CRC. The mean OS duration after surgery was 69.50 months (95% confidence interval [CI]: 65.24–73.75; Fig 3A). Kaplan–Meier and univariate regression analyses revealed that an advanced T stage (III–IV vs. I–II; hazard ratio [HR]: 2.015, 95% CI: 1.197–3.390; $P = 0.008$; Fig 3H), N stage (N1–N2 vs. N0; HR: 3.839, 95% CI: 2.181–6.757; $P < 0.001$; Fig 3I), M stage (M1 vs. M0; HR: 3.788, 95% CI: 2.026–7.081; $P < 0.001$; Fig 3J), TNM stage (I–II vs. III-IV; HR: 3.832, 95% CI: 2.220-6.614; $P < 0.001$; Fig 3K), and FAM50A expression level (high vs. low; HR: 2.544, 95% CI: 1.504–4.303; $P < 0.001$; Fig 3L) were associated with poor OS. There

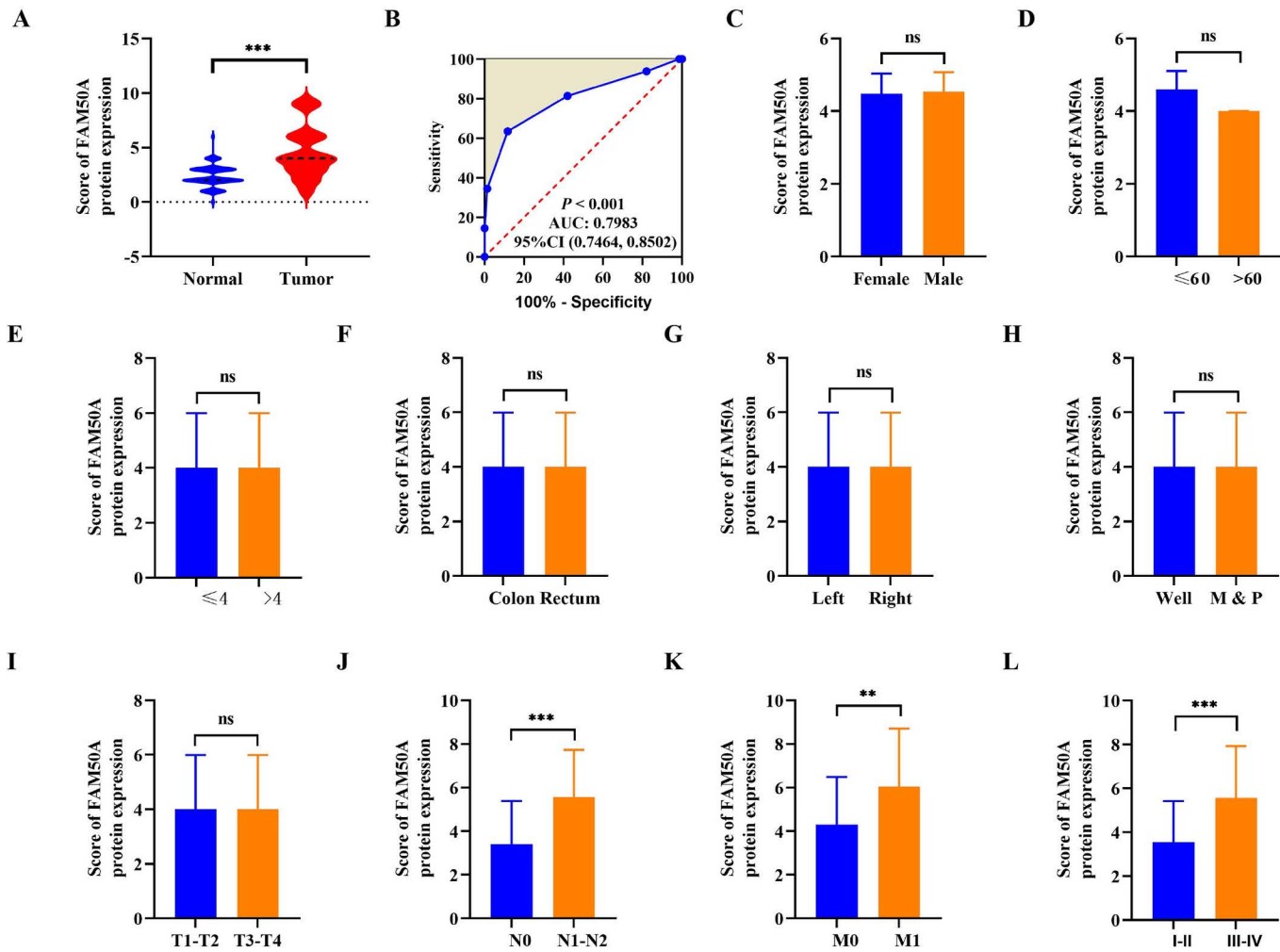

**Fig 2. Correlation of FAM50A expression with clinical and pathological features of CRC.** (A) Expression of FAM50A in normal and cancer tissues. (B) ROC curve analysis identified the different expression between the two groups. (C) Expression of FAM50A in male and female patients with CRC. (D) Expression of FAM50A in different age groups. (E) Expression of FAM50A in relation to tumor size. (F) Expression of FAM50A in relation to tumor location. (G) FAM50A expression in relation to cancer site. (H) FAM50A expression in relation to histological differentiation (M, moderate differentiation; P, poor differentiation). (I) FAM50A expression in relation to the depth of tumor invasion. (J) FAM50A expression according to lymph node metastasis. (K) FAM50A expression according to the presence of distant metastasis. (L) FAM50A expression according to TNM stage. * $P \leq 0.05$, ** $P \leq 0.01$, *** $P \leq 0.001$, ns: not significant.

was no statistically significant difference in other variables (Figs 3B-3G, all $P > 0.05$). The corresponding risk factors are presented in Table 3. Multivariate Cox regression analysis showed that advanced M stage, and high FAM50A expression level were independent predictors of poor OS (both $P < 0.05$, Table 3). These findings indicated that FAM50A may be a prognostic biomarker in predicting the survival outcome of patients with CRC.

Next, forest plots were drawn to visually illustrate the prognostic impact of FAM50A. These revealed a significant association between high FAM50A expression levels and the poor survival of patients with CRC (Figs 4A and 4B, Table 3). Nomograms can employ multiple indicators to simultaneously diagnose or predict the onset or progression of disease [28]. In the present study, a nomogram was constructed to display the prognostic ability of FAM50A levels in patients with CRC. Candidate prognostic variables were assessed using a proportional

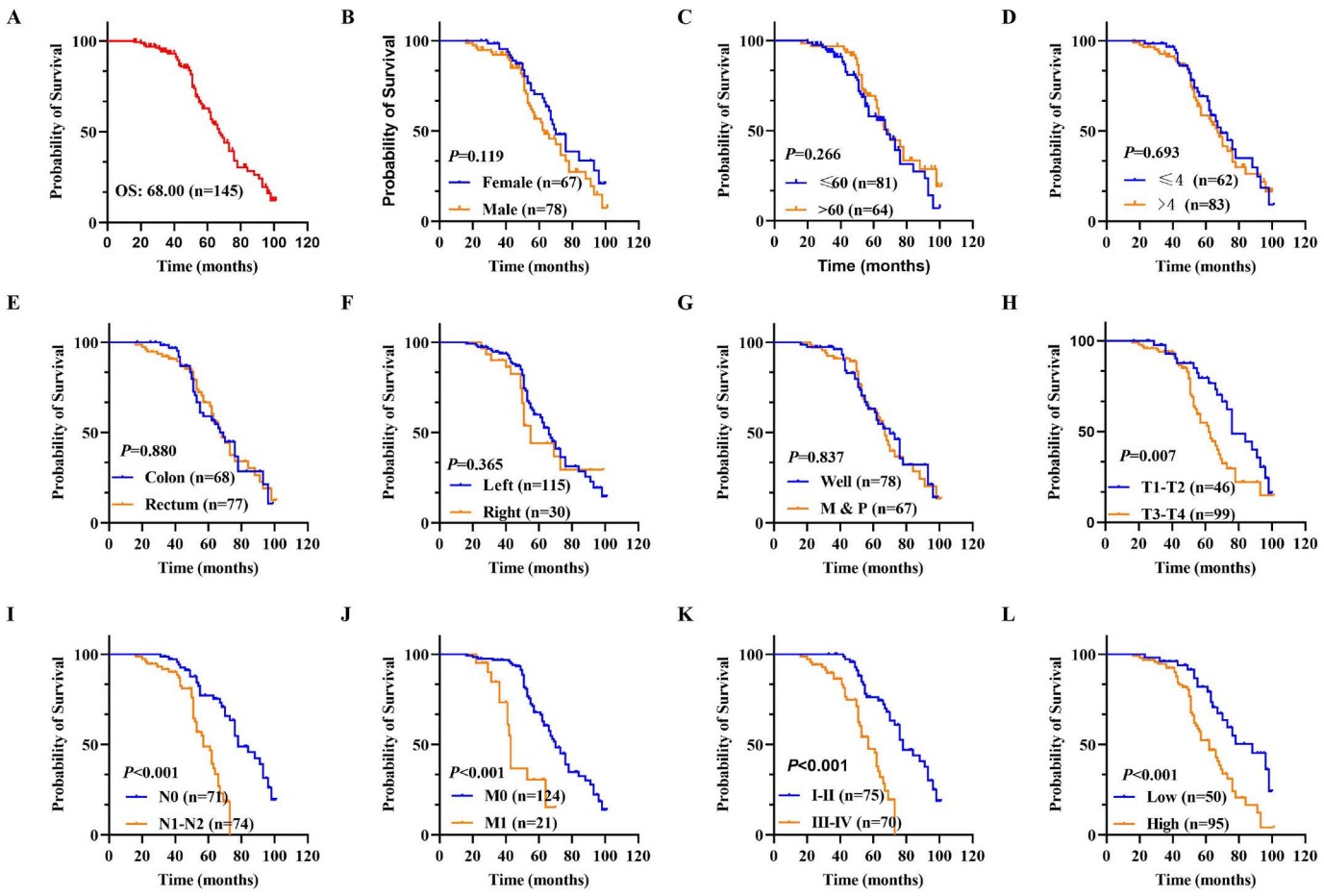

**Fig 3. Cox regression analysis in CRC.** (A) Overall survival rate in CRC patients (n = 145). (B) OS in female and male groups. (C) OS in different age groups. (D) OS related to tumor size. (E) OS related to tumor location. (F) OS of ddifferent cancer sites. (G) OS interrelated with differentiation. (H-K) OS in T, N, M and TNM stages. (L) OS in FAM50A high and low groups. *$P \leq$ 0.05, **$P \leq$ 0.01, ***$P \leq$ 0.001 and ns: no significance.

hazard plot. This revealed that T, N, M, and TNM stage, as well as FAM50A expression level, all fitted the inclusion criteria (Figs 5A–5E). A nomogram was subsequently constructed using these five variables and subsequently assessed using a calibration curve. A high level of FAM50A expression was associated with higher scores and with a decreased probability of patient survival (Fig 5F). The calibration curve also indicated that the nomogram was well-constructed, since the predicted values were in good agreement with the actual values (Fig 5G).

### 3.4. Construction of cell lines with stable FAM50A expression

Continuous proliferation signals and the evasion of growth inhibition are characteristic phenotypes of tumor cells. We, therefore, investigated the potential role of FAM50A in promoting tumor cell proliferation. The gene expression level of *FAM50A* was first analyzed in four CRC cell lines (SW480, HCT-8, HCT-116, and RKO) and in normal intestinal epithelial cells (FHC). Results from RT-qPCR and western blotting analyses revealed higher FAM50A expression levels in tumor cell lines than normal FHC cells, at both the mRNA (Fig 6A, upper panel) and protein levels (Fig 6A, lower panel). The SW480 and HCT-8 cell lines were used

**Table 3. Univariate and multivariate Cox proportional hazards regression analysis in CRC patients.**

| Clinicopathologic parameters | | Median of OS (95% CI) | 5-year OS (%) | Univariate analysis | | Multivariate analysis | |
|---|---|---|---|---|---|---|---|
| | | | | HR (95% CI) | *P*-value | HR (95% CI) | *P*-value |
| Total | | 68.00 (62.48–73.52) | 63.00 | | | | |
| Gender | Female | 70.00 (62.54–77.46) | 70.00 | 1.437 (0.904–2.285) | 0.125 | | |
| | Male | 63.47 (51.56–75.38) | 56.70 | | | | |
| Age | ≤60 | 68.00 (57.85–78.15) | 57.70 | 0.772 (0.486–1.225) | 0.272 | | |
| | >60 | 69.00 (61.10–76.90) | 69.20 | | | | |
| Tumor size | ≤4 cm | 69.00 (57.79–80.20) | 69.00 | 1.097 (0.689–1.746) | 0.696 | | |
| | >4 cm | 67.00 (60.61–73.39) | 56.80 | | | | |
| Tumor location | Colon | 67.00 (57.94–76.05) | 58.70 | 0.966 (0.610–1.527) | 0.881 | | |
| | Rectum | 67.00 (59.11–74.89) | 66.60 | | | | |
| Cancer site | Left | 66.00 (59.85–72.15) | 58.70 | 0.775 (0.443–1.356) | 0.775 | | |
| | Right | 76.00 (66.90–85.10) | 70.20 | | | | |
| Differentiation | Well | 70.00 (61.09–78.91) | 64.90 | 1.049(0.663–1.658) | 0.839 | | |
| | M-P | 67.00 (61.63–72.38) | 62.60 | | | | |
| Depth of tumor invasion | T1-T2 | 76.00 (64.38–87.62) | 79.50 | 2.015 (1.197–3.390) | 0.008 | 1.564 (0.912–2.680) | 0.104 |
| | T3-T4 | 62.18 (55.22–69.14) | 53.00 | | | | |
| Lymph node metastasis | N0 | 78.00 (67.28–88.72) | 77.00 | 3.839 (2.181–6.757) | < 0.001 | 1.505 (0.445–5.085) | 0.511 |
| | N1-N2 | 56.92 (47.95–65.89) | 48.50 | | | | |
| Distant metastasis | M0 | 70.00 (63.59–76.41) | 67.60 | 3.788 (2.026–7.081) | < 0.001 | 3.137 (1.543–6.337) | 0.002 |
| | M1 | 42.83 (41.02–44.64) | 30.60 | | | | |
| TNM stage | I-II | 78.00 (68.39–87.61) | 76.10 | 3.832 (2.220–6.614) | < 0.001 | 1.565 (0.468–5.229) | 0.467 |
| | III-IV | 56.69 (49.15–64.19) | 47.70 | | | | |
| FAM50A | Low | 87.81 (68.99–106.62) | 79.00 | 2.544 (1.504–4.303) | < 0.001 | 2.269 (1.304–3.946) | 0.004 |
| | High | 62.08 (53.46–70.69) | 52.20 | | | | |

to construct knockdown cell lines as they revealed the highest expression levels. Furthermore, the expression level of *FAM50A* in RKO cells was relatively lower and therefore, these cells were used for overexpression. Cells were first transfected with lentivirus, followed by drug screening to obtain cells with stable knockdown or overexpression [37]. These stably expressing cells were then used in experiments to study the cell phenotype, as described below.

### 3.5. FAM50A promoted CRC cell proliferation

The efficiencies of knockdown were verified at the mRNA and protein level in SW480 (Fig 6B), and HCT-8 (Fig 6C) cells. These stably expressing cells were then used in experiments to study the cell phenotype, as described below. The proliferation of CRC cells was evaluated using the CCK-8 method. Knockdown of FAM50A significantly reduced the proliferation of SW480 (Fig 6D) and HCT-8 (Fig 6E) cells. EdU staining was used to evaluate the effect of FAM50A knockdown on cell proliferation ability. Knockdown of FAM50A significantly reduced the proportion of EdU-positive sh-FAM50A-1 and sh-FAM50A-2 cells (Fig 6F). In addition, the number of cell clones was greatly reduced in cells with low FAM50A expression levels (Fig 6G). Moreover, overexpression of FAM50A was verified (Fig 7A), and significantly enhanced the proliferation ability of the RKO cell line (Fig 7B). The results of the EdU staining and colony-formation experiments demonstrated that overexpression of FAM50A significantly increased the number of EdU-positive cells (Fig 7C) and the extent of colony formation (Fig 7D). Overall, these results confirmed the role of FAM50A in cell proliferation.

A

| Characteristics | Total(N) | HR (95% CI) | | P value |
|---|---|---|---|---|
| Gender | | | | |
| Female | 67 | | | |
| Male | 78 | 1.437 (0.904-2.285) | | 0.125 |
| Age | | | | |
| ≤60 | 81 | | | |
| >60 | 64 | 0.772 (0.486-1.225) | | 0.272 |
| Tumor size | | | | |
| ≤4cm | 62 | | | |
| >4cm | 83 | 1.097 (0.689-1.746) | | 0.696 |
| Tumor location | | | | |
| Colon | 68 | | | |
| Rectum | 77 | 0.966 (0.610-1.527) | | 0.881 |
| Cancer site | | | | |
| Left | 115 | | | |
| Right | 30 | 0.775 (0.443-1.356) | | 0.775 |
| Differentiation | | | | |
| Well | 78 | | | |
| Moderate-Poor | 67 | 1.049 (0.663-1.658) | | 0.839 |
| Depth of tumor invasion | | | | |
| T1-T2 | 46 | | | |
| T3-T4 | 99 | 2.015 (1.197-3.390) | | 0.008 |
| Lymph node metastasis | | | | |
| N0 | 71 | | | |
| N1-N2 | 74 | 3.839 (2.181-6.757) | | < 0.001 |
| Distant metastasis | | | | |
| M0 | 124 | | | |
| M1 | 21 | 3.788 (2.026-7.081) | | < 0.001 |
| TNM stage | | | | |
| I-II | 75 | | | |
| III-IV | 70 | 3.832 (2.220-6.614) | | < 0.001 |
| FAM50A | | | | |
| Low | 50 | | | |
| High | 95 | 2.544 (1.504-4.303) | | < 0.001 |

B

| Characteristics | Total(N) | HR (95% CI) | | P value |
|---|---|---|---|---|
| Depth of tumor invasion | | | | |
| T1-T2 | 46 | | | |
| T3-T4 | 99 | 1.564 (0.912-2.680) | | 0.104 |
| Lymph node metastasis | | | | |
| N0 | 71 | | | |
| N1-N2 | 74 | 1.505 (0.445-5.085) | | 0.511 |
| Distant metastasis | | | | |
| M0 | 124 | | | |
| M1 | 21 | 3.137 (1.543-6.337) | | 0.002 |
| TNM stage | | | | |
| I-II | 75 | | | |
| III-IV | 70 | 1.565 (0.468-5.229) | | 0.467 |
| FAM50A | | | | |
| Low | 50 | | | |
| High | 95 | 2.269 (1.304-3.946) | | 0.004 |

**Fig 4. Forest plot of prognostic risk factors for CRC.** (A) Univariate Cox regression analysis forest plot. (B) Multivariate Cox regression analysis forest plot.

## 3.6. FAM50A promotes the proliferation of CRC cells by affecting cell cycle and the expression of cell cycle-related proteins

The abnormality of the cell cycle is closely related to the occurrence and development of tumors [38]. To explore the probable mechanism by which FAM50A is involved in tumor progression, the changes in the cell cycle were examined by using flow cytometry. The results of flow cytometry showed that significant changes in the proportion of cells at different stages of the cell cycle. After knocking down FAM50A, there was a significant increase in the number of cells in the S phase, indicating the occurrence of S phase arrest (sh-FAM50A vs. sh-ctrl, $P < 0.001$, Fig 8A: in SW480, Fig 8B: in HCT-8). These results suggested that knockdown of FAM50A indeed leaded to cell cycle arrest. Given that cell cycle

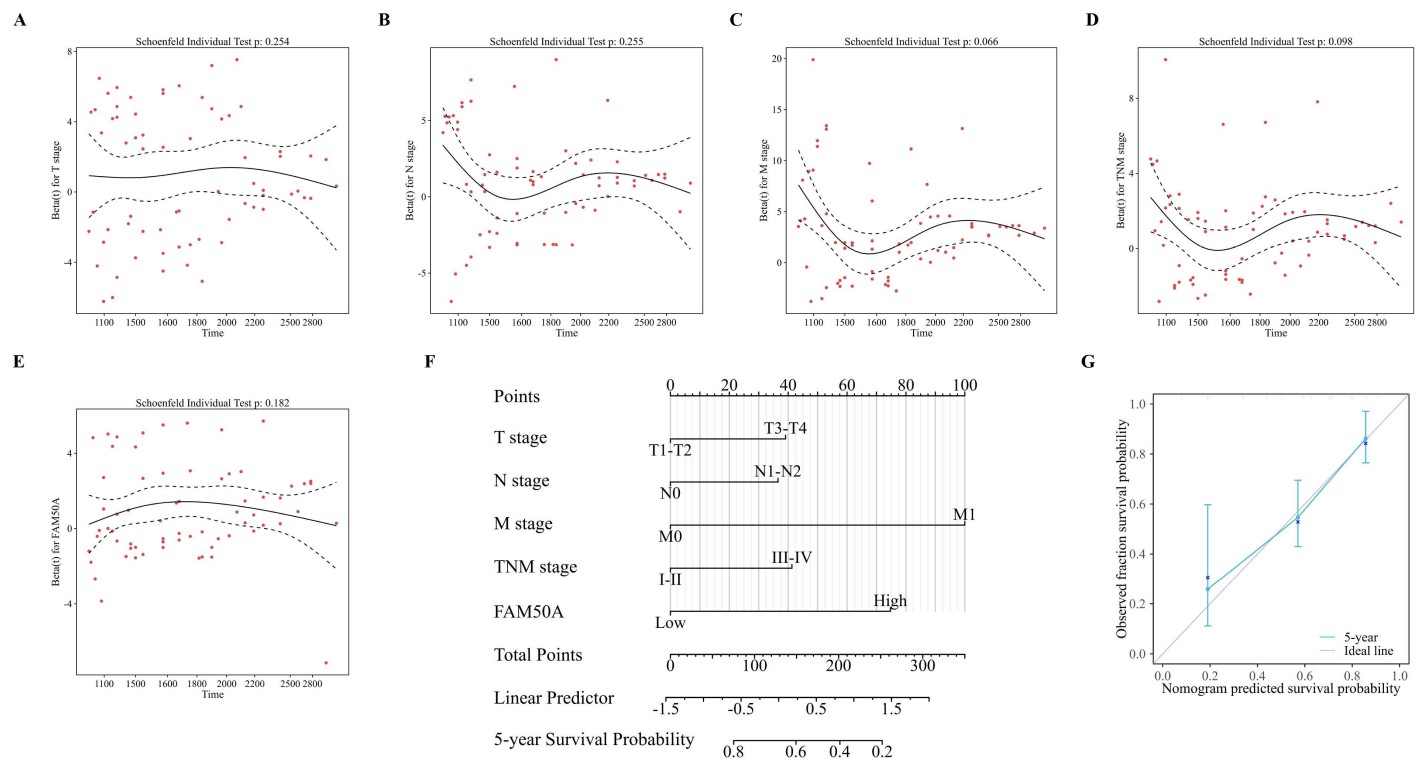

**Fig 5. Nomogram to display the prognosis value of FAM50A in CRC.** (A-E) represent the results of prognostic proportional risk analysis in T, N, M and TNM stages and FM50A expression. (F) Nomogram constructed with different variables. (G) Calibration curve of nomogram.

proteins and cyclin-dependent kinases play crucial roles in the process of cell proliferation and can drive cells to progress from one stage to the next [39], these proteins were tested one by one. No surprisingly, CyclinA2 and CDK2 were significantly reduced in FAM50A knocking down cells (Figs 8C and 8D). These findings indicated that FAM50A may be involved in the malignant progression of CRC through interactions with cell cycle and cell cycle-related proteins.

## 4. Discussion

Ongoing research in CRC has demonstrated that for effective diagnosis and treatment, potential targets must exhibit significant differential expression at the mRNA or protein level between tumor and corresponding normal tissue. Additionally, the target should demonstrate specific functional effects in terms of promoting or suppressing tumor growth [40]. We observed that the FAM50A expression level was significantly elevated in tumor tissues compared to adjacent normal tissues, suggesting that it may serve as a potential biomarker in CRC. The clinical significance of genes such as *p53*, *C-myc*, *Ras*, *EGFR*, *PMS1*, *PMS2*, *COX-2*, *CD44*, *PD-1*, *PD-L1*, and *CTLA4* in cancer has been extensively investigated. This includes the study of inactivated tumor suppressor genes (*p53*), activated proto-oncogenes (*C-myc*, *Ras*, and *EGFR*), mismatch repair genes (*PMS1* and *PMS2*), modifying genes (*COX-2* and *CD44*), and immune-related genes (*PD-1*, *PD-L1*, and *CTLA4*). These genes play crucial roles in genetic diagnosis, prognostic assessment, immunotherapy, and targeted therapy [41–45]. The results of the current study revealed significant correlations between FAM50A expression levels and the clinical features of N, M, and

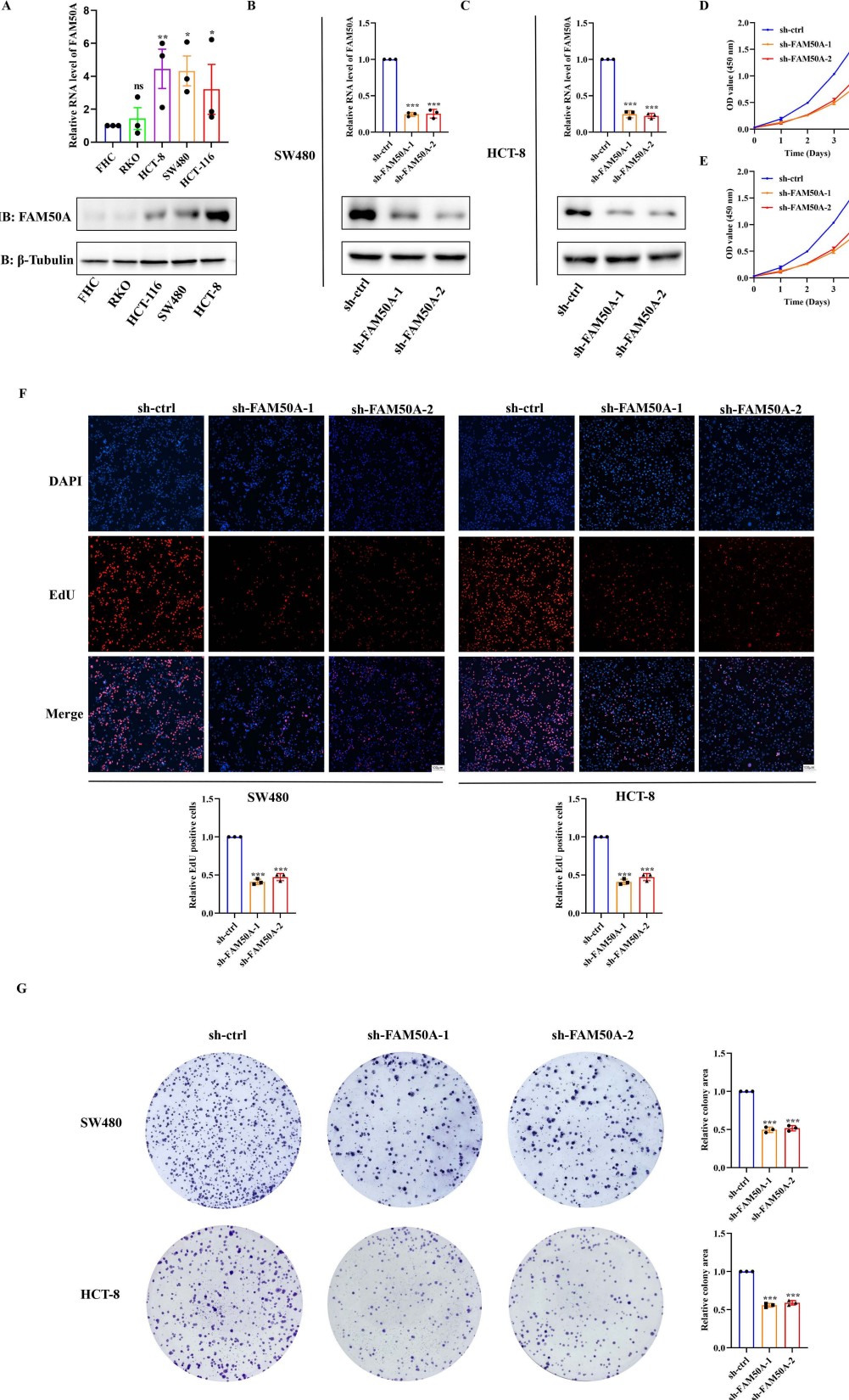

**Fig 6. Knockdown of FAM50A inhibited CRC cells proliferation ability in vitro.** (A) Detection of FAM50A expression in CRC tumor cell lines and normal cell by using RT-qPCR and WB. (B) Verification of knockdown efficiency in

SW480. (C) Verification of knockdown efficiency in HCT-8. (D) Detection of cell proliferation ability by CCK-8 assays in SW480. (E) CCK-8 assays in HCT-8. (F) EdU staining to identify the changes in cell proliferation after FAM50A knockdown in SW480 and HCT-8. (G) Plate colony formation assay to identify changes in cell proliferation in FAM50A knockdown cells. sh means knockdown, $*P \leq 0.05$, $**P \leq 0.01$, $***P \leq 0.001$ and ns: no significance.

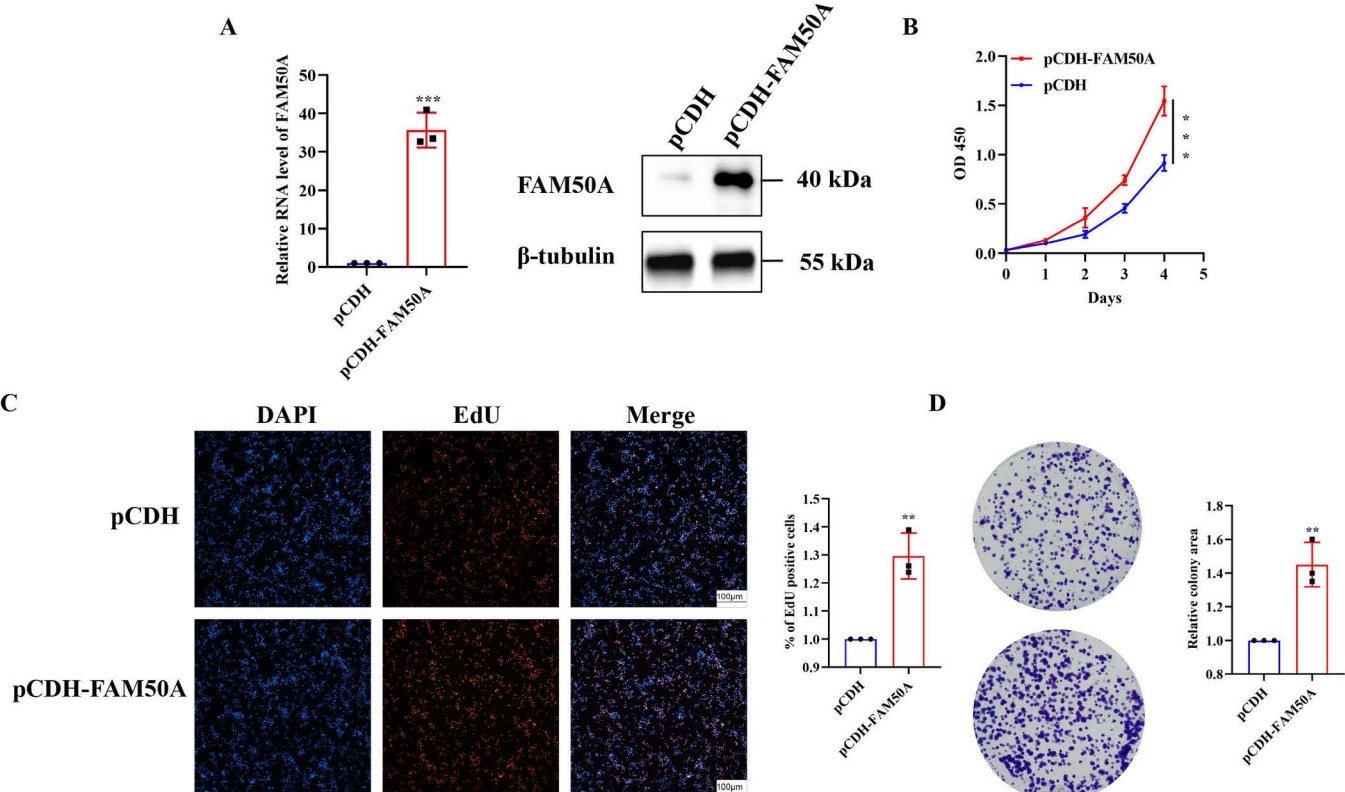

**Fig 7. Overexpression of FAM50A promoted cell proliferation ability in RKO.** (A) Evaluation of the high expression efficiency for FAM50A. (B) CCK-8 assays in RKO. (C) EdU stained cells in overexpression cell compared with control RKO cell. (D) The colony forming ability assay of RKO.

TNM stage, which is consistent with the results of a related study [46]. Together, they suggest that FAM50A may be a valuable prognostic marker for clinical use. Multivariate Cox regression analysis demonstrated that the FAM50A expression level was an independent prognostic factor in CRC.

There is limited research on FAM50A, and its function has yet to be fully elucidated. The expression level of FAM50A varies across different tissues and tumors in the human body, as revealed by pan-cancer analysis of public databases. This variability may be attributed to tissue specificity, the involvement of diverse signaling regulatory mechanisms, and distinct epigenetic modifications [47]. The results for FAM50A expression in the TCGA database were validated by IHC, thus highlighting its potential oncogenic activity. To further investigate its potential role as an oncogene, we constructed HCT-8 and SW480 cell lines with reduced FAM50A expression levels and RKO with with overexpressed FAM50A, then confirmed the efficacy using qRT-PCR and western blotting methods. Given the importance of tumor cell proliferation [48], our initial focus was to examine the effect of FAM50A on the proliferation of malignant CRC cells. CCK-8, EdU, and colony-forming assays were performed to

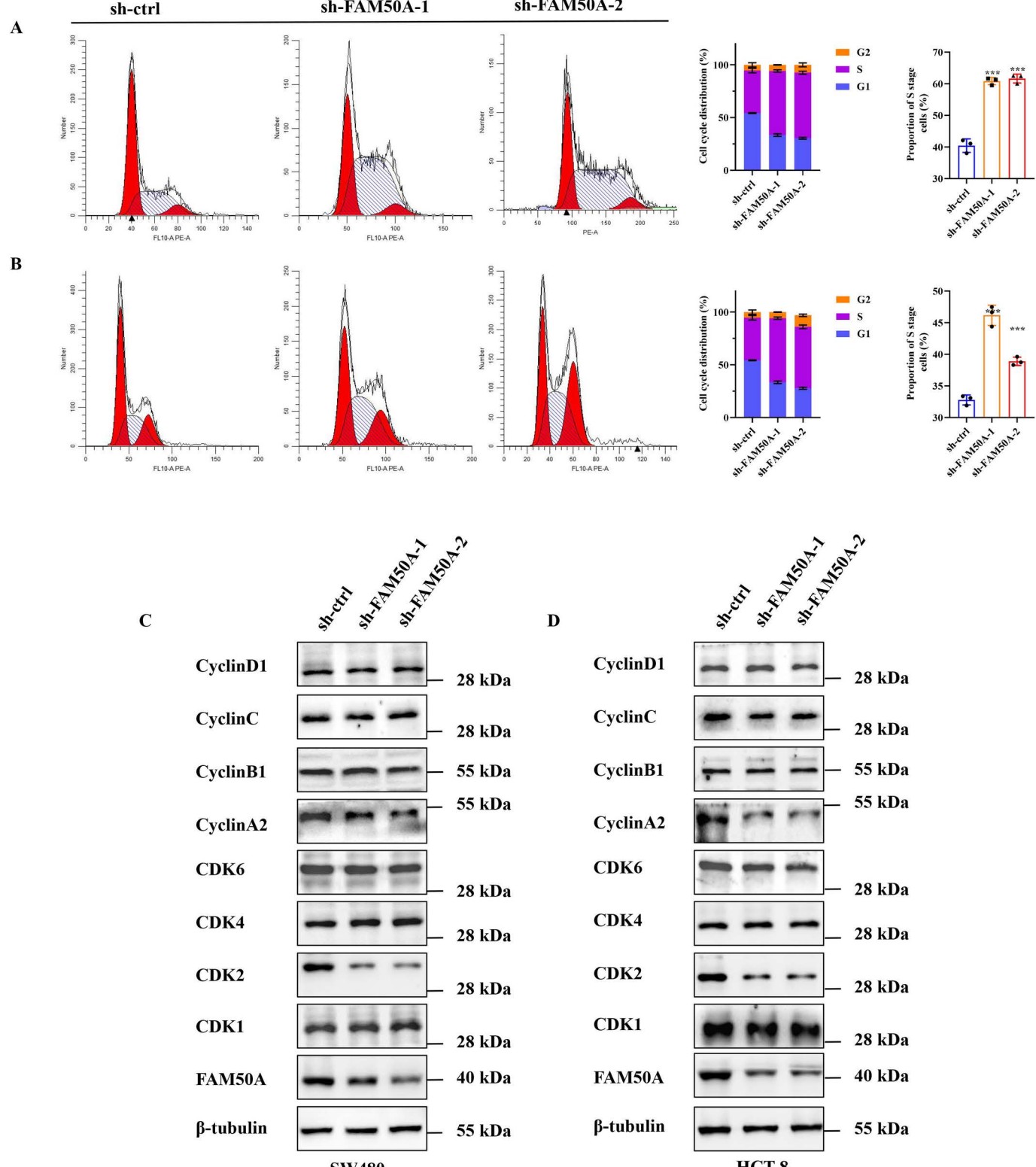

**Fig 8. Knocking down FAM50A leads to S phase arrest in CRC cells.** (A-B) Cell cycle assays were performed on SW480 and HCT-8 cells after FAM50A was knocked down (up: SW480, down: HCT-8). (C-D) Western blot assays were conducted to detect the expression of cell cycle and cell cycle-related proteins in CRC cells.

study the effects of FAM50A, with the results consistently demonstrating the impacts on cell proliferation.

The FAM family members perform important functions in the proliferation, apoptosis, invasion and metastasis of tumor cells, as well as in the regulation of the tumor microenvironment [49,50]. This study specifically focused on the role of FAM50A in tumor proliferation and found that it may be linked to CyclinA2 and CDK2. As a study has shown, a FAM family member, FAM84B, regulates the proliferation of glioma cells by acting on the cell cycle pathways like CDK2 [51]. In tumor cells such as those of breast cancer and lung cancer, CDK2 is often hyperactivated. It propels cells from the G1 phase to the S phase and promotes DNA synthesis, thereby leading to abnormal proliferation of tumor cells [52,53]. Abnormal expression of CyclinA2 can be detected in numerous tumors. The CyclinA2/CDK complex phosphorylates multiple substrate proteins and regulates key cellular processes such as DNA replication and chromosome segregation, providing a driving force for the continuous growth of tumor cells and enabling their rapid proliferation [54,55]. A research has revealed that when FAM50A was knocked down, it could lead to DNA damage, elicit the expression of interferon beta and interleukin-6, and inhibit the proliferation, invasion and migration of cancer cells [56], which is basically in accordance with our results in this study. Based on this previous study and our current findings, we hypothesize that FAM50A may play roles in CRC by targeting the CyclinA2/CDK2 signal pathway.

While our findings demonstrated the prognostic value of FAM50A in CRC and a possible role in promoting tumor cell proliferation, several issues warrant further exploration. First, the limited sample size and pathology information of our CRC patient cohort may have affected the reliability and generalizability of our results. Samples with relevant patient information will be collected for analysis in future research endeavors. Additionally, the cell lines used for *in vitro* studies may not replicate the intricate and diverse nature of human CRC because of the lack of appropriate immune and physiological environments. Therefore, we will consider incorporating further *in vitro* testing in more comprehensive investigations in future studies. Financial constraints also restricted the scale of the experiments, the number of replicates, and the use of more sophisticated techniques. If various factors, such as funding and time, permit, our team Further *in vivo* research will conduct more experiments in vivo, which could may provide more precise results and conclusions. Importantly, the specific mechanism involving whereby FAM50A in contributed to CRC progression has not been fully elucidated and requires more in-depth study.

In summary, our results revealed that FAM50A may be a novel biomarker for CRC prognosis; and moreover, FAM50A may participate in regulating tumor progression by targeting CyclinA2/CDK pathway to drive CRC cell proliferation.

## Supporting information

**S1 File. Raw images.pdf.**
(ZIP)

## Acknowledgments

We would like to express our sincere gratitude to the relevant data provided by the public database.

## Author contributions

**Conceptualization:** Long-Hai Li, Hao Gu, Yan Sun.

**Data curation:** Long-Hai Li, Guangyun Li.

**Formal analysis:** Hao Gu.

**Funding acquisition:** Long-Hai Li.

**Investigation:** Zhaoshuai Ji, Guangyun Li.

**Project administration:** Zhaoshuai Ji.

**Supervision:** Yan Sun.

**Visualization:** Zhaoshuai Ji.

**Writing – original draft:** Yan Sun.

**Writing – review & editing:** Long-hai Li.

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
