## [Decision Letter · Decision Letter 0]

10 Dec 2024

PONE-D-24-50052FAM50A acts as a novel prognostic marker and modulates the proliferation of colorectal cancer cells via CylinA2/CDK2 pathwayPLOS ONE

Dear Dr. Li,

Thank you for submitting your manuscript to PLOS ONE. After careful consideration, we feel that it has merit but does not fully meet PLOS ONE’s publication criteria as it currently stands. Therefore, we invite you to submit a revised version of the manuscript that addresses the points raised during the review process.

We look forward to receiving your revised manuscript.

Kind regards,

Peng Zhang, Ph.D.

Academic Editor

PLOS ONE

Journal requirements: When submitting your revision, we need you to address these additional requirements. 1. Please ensure that your manuscript meets PLOS ONE's style requirements, including those for file naming. The PLOS ONE style templates can be found at https://journals.plos.org/plosone/s/file?id=wjVg/PLOSOne_formatting_sample_main_body.pdf and https://journals.plos.org/plosone/s/file?id=ba62/PLOSOne_formatting_sample_title_authors_affiliations.pdf 2. PLOS ONE now requires that authors provide the original uncropped and unadjusted images underlying all blot or gel results reported in a submission’s figures or Supporting Information files. This policy and the journal’s other requirements for blot/gel reporting and figure preparation are described in detail at https://journals.plos.org/plosone/s/figures#loc-blot-and-gel-reporting-requirements and https://journals.plos.org/plosone/s/figures#loc-preparing-figures-from-image-files. When you submit your revised manuscript, please ensure that your figures adhere fully to these guidelines and provide the original underlying images for all blot or gel data reported in your submission. See the following link for instructions on providing the original image data: https://journals.plos.org/plosone/s/figures#loc-original-images-for-blots-and-gels.   In your cover letter, please note whether your blot/gel image data are in Supporting Information or posted at a public data repository, provide the repository URL if relevant, and provide specific details as to which raw blot/gel images, if any, are not available. Email us at plosone@plos.org if you have any questions. 3. Thank you for stating the following financial disclosure:  [ the Bozhou Key R&D projects (No. bzzc2021020)].  Please state what role the funders took in the study.  If the funders had no role, please state: ""The funders had no role in study design, data collection and analysis, decision to publish, or preparation of the manuscript."" If this statement is not correct you must amend it as needed. Please include this amended Role of Funder statement in your cover letter; we will change the online submission form on your behalf. 4. In the online submission form, you indicated that [The data underlying the results presented in the study are available from the authors.]. All PLOS journals now require all data underlying the findings described in their manuscript to be freely available to other researchers, either 1. In a public repository, 2. Within the manuscript itself, or 3. Uploaded as supplementary information.This policy applies to all data except where public deposition would breach compliance with the protocol approved by your research ethics board. If your data cannot be made publicly available for ethical or legal reasons (e.g., public availability would compromise patient privacy), please explain your reasons on resubmission and your exemption request will be escalated for approval. 5. Please include captions for your Supporting Information files at the end of your manuscript, and update any in-text citations to match accordingly. Please see our Supporting Information guidelines for more information: http://journals.plos.org/plosone/s/supporting-information. 

Reviewers' comments:

Reviewer's Responses to Questions

**Comments to the Author**

1. Is the manuscript technically sound, and do the data support the conclusions?

Reviewer #1: Yes

Reviewer #2: Yes

Reviewer #3: Yes

2. Has the statistical analysis been performed appropriately and rigorously? 

Reviewer #1: Yes

Reviewer #2: Yes

Reviewer #3: Yes

3. Have the authors made all data underlying the findings in their manuscript fully available?

Reviewer #1: Yes

Reviewer #2: Yes

Reviewer #3: Yes

4. Is the manuscript presented in an intelligible fashion and written in standard English?

Reviewer #1: Yes

Reviewer #2: Yes

Reviewer #3: Yes

5. Review Comments to the Author

Reviewer #1: This study identified FAM50A as a potential novel prognostic marker for colorectal cancer (CRC) and, for the first time, clarified its role in regulating CRC cell proliferation via the CyclinA2/CDK2 pathway. This offers a new perspective and potential target for the diagnosis and treatment of CRC. This article suggests that FAM50A may promote tumor cell proliferation via the CyclinA2/CDK2 signaling pathway, the direct interaction for this mechanism needs to be improved, as least some interaction models.

Specific issues in the text should be clarified:

1.In the text, the 252nd line mentions ‘Using ROC curve analysis’This should be reflected in the figure with the inclusion of an ROC curve graph.

2.In academic papers, it is crucial to provide clear and concise figure legends to assist readers in quickly comprehending the content of the figures. The subplot labels ABCD in Figure 6 and 7 are inconsistent with other figure formats and should be standardized.

3.The figure 6 caption has a mismatch between parts F and G, and it is suggested that a statistical analysis should be conducted on the 'Edu' part to make it more concise.

4.It is suggested that a flowchart be added to Figures A and B in Figure 8 to enhance the persuasiveness of the evidence.

5.Clearly label the samples on the WB (Western blot) lanes to visually compare the detection results of each lane in Figure 8.

6.Figure 3 and Figure 6 F should include the "n" value in the statistical graphs to ensure the transparency, reproducibility, and reliability of the research results.

Reviewer #2: I have reviewed the manuscript titled “FAM50A acts as a novel prognostic marker and modulates the proliferation of colorectal cancer cells via CylinA2/CDK2 pathway”. The authors have investigated FAM50A across different types of cancer and then functionally evaluated the gene in CRC by various experimental methods. The manuscript is well designed and the results are valuable. However, there are several concerns that need to be addressed in this study.

General comments

-The title could be changed to “FAM50A as a novel prognostic marker modulates the proliferation of colorectal cancer cells via CylinA2/CDK2 pathway”. The suggested title is shorter and more precise.

-What was the rational to evaluate just FAM50A and the authors didn’t investigate FAM50B as well?

-There is one article titled “Identification and Clinical Validation of a Novel 4 Gene-Signature with Prognostic Utility in Colorectal Cancer” that pointed to the significance of FAM50A in CRC. So, it could be referenced. Also, the authors could talk about the article in the introduction and discussion.

-There are typo/grammar errors that should be revised (i.e. Finally, the changes in the cell cycle following FAM50A knockdown were detected by using flow cytometry; using could be omitted).

Abstract

Since the gene expression and prognostic analysis were performed by TCGA and GTEX, its better to mention the databases in the abstract. In the current form it seem that the authors evaluated these variables by experimental analysis in their labs.

Materials and Methods

-The primers sequences should be provided as a table.

Reviewer #3: The artice is devoted to the actual topic concerning the search of a new prognostic marker for colorectal cancer. The article is clearly written and contains a sufficient amount of introduction and discussion. The results are well structured and illustrated. I recommend the article for acceptance.

6. PLOS authors have the option to publish the peer review history of their article (what does this mean? ). If published, this will include your full peer review and any attached files.

**Do you want your identity to be public for this peer review?** For information about this choice, including consent withdrawal, please see our Privacy Policy .

Reviewer #1: No

Reviewer #2: **Yes: ** Zahra Salehi

Reviewer #3: No

---

## [Author Response · Author response to Decision Letter 0]

9 Jan 2025

To editor:

Thank you very much. We have completed the issues you raised and made the corresponding revisions.

To reviewer:

Reviewer #1: This study identified FAM50A as a potential novel prognostic marker for colorectal cancer (CRC) and, for the first time, clarified its role in regulating CRC cell proliferation via the CyclinA2/CDK2 pathway. This offers a new perspective and potential target for the diagnosis and treatment of CRC. This article suggests that FAM50A may promote tumor cell proliferation via the CyclinA2/CDK2 signaling pathway, the direct interaction for this mechanism needs to be improved, as least some interaction models.

Specific issues in the text should be clarified:

1.In the text, the 252nd line mentions ‘Using ROC curve analysis’This should be reflected in the figure with the inclusion of an ROC curve graph.

ANSWER: With reference to your valuable suggestions, we have recalculated the optimal cut-off value of the scores and drawn the ROC curve in Figure 2A, 2B. And we have also reorganized the layout of Figure 2.

2.In academic papers, it is crucial to provide clear and concise figure legends to assist readers in quickly comprehending the content of the figures. The subplot labels ABCD in Figure 6 and 7 are inconsistent with other figure formats and should be standardized.

ANSWER: We have already made the revisions.

3.The figure 6 caption has a mismatch between parts F and G, and it is suggested that a statistical analysis should be conducted on the 'Edu' part to make it more concise.

ANSWER: We have already added the statistical description of the EdU results and adjusted the positions of F and G at the same time.

4.It is suggested that a flowchart be added to Figures A and B in Figure 8 to enhance the persuasiveness of the evidence.

ANSWER: Referring to your suggestions, we have added the flow cytometry result images.

5.Clearly label the samples on the WB (Western blot) lanes to visually compare the detection results of each lane in Figure 8.

ANSWER: We have already added the labels for the samples of WB lane .

6. Figure 3 and Figure 6 F should include the "n" value in the statistical graphs to ensure the transparency, reproducibility, and reliability of the research results.

ANSWER: Thank you very much for your suggestions. With reference to your advice, we have revised the figures in the article as much as possible to show the "n" value in the statistical graphs.

Reviewer #2: I have reviewed the manuscript titled “FAM50A acts as a novel prognostic marker and modulates the proliferation of colorectal cancer cells via CylinA2/CDK2 pathway”. The authors have investigated FAM50A across different types of cancer and then functionally evaluated the gene in CRC by various experimental methods. The manuscript is well designed and the results are valuable. However, there are several concerns that need to be addressed in this study.

General comments

1.-The title could be changed to “FAM50A as a novel prognostic marker modulates the proliferation of colorectal cancer cells via CylinA2/CDK2 pathway”. The suggested title is shorter and more precise.

2.-What was the rational to evaluate just FAM50A and the authors didn’t investigate FAM50B as well?

3. -There is one article titled “Identification and Clinical Validation of a Novel 4 Gene-Signature with Prognostic Utility in Colorectal Cancer” that pointed to the significance of FAM50A in CRC. So, it could be referenced. Also, the authors could talk about the article in the introduction and discussion.

4. -There are typo/grammar errors that should be revised (i.e. Finally, the changes in the cell cycle following FAM50A knockdown were detected by using flow cytometry; using could be omitted).

ANSWER：

1. Referring to your valuable suggestions, we have revised the title in the article.

2.In our research, the main reason why we focused on FAM50A and didn't conduct an in-depth investigation into FAM50B is that significant differences in expression patterns were found during the detection of tumor tissue samples. The expression level of FAM50A in cancer tissues within tumors is higher than that in adjacent non-cancerous tissues, while the expression level of FAM50B in tumor tissues is lower than that in adjacent non-cancerous tissues. Such an expression characteristic of FAM50A often coincides with the behavioral manifestations of oncogenes. Moreover, our research team has been committed to the field of oncogene-related research for a long time and has a strong interest in and profound research experience regarding molecules with potential oncogene properties. Therefore, we have placed the research focus on FAM50A this time. However, we will also pay attention to the related roles of FAM50B in the future research process.

3. Based on your suggestions, we have read this literature carefully and added a discussion in the article. (Line 105, 107-108, Reference 22)

4. Referring to your suggestions, we have made grammar modifications and checked again.

Abstract

Since the gene expression and prognostic analysis were performed by TCGA and GTEX, its better to mention the databases in the abstract. In the current form it seem that the authors evaluated these variables by experimental analysis in their labs.

ANSWER：Referring to your valuable suggestions, we have mentioned the use of TCGA and GTEX database in the abstract.

Materials and Methods

-The primers sequences should be provided as a table.

ANSWER：Referring to your suggestions, we have already put the primers into the table.

Reviewer #3: The artice is devoted to the actual topic concerning the search of a new prognostic marker for colorectal cancer. The article is clearly written and contains a sufficient amount of introduction and discussion. The results are well structured and illustrated. I recommend the article for acceptance.

同意接收出版。

ANSWER：Thank you very much.

---

## [Editor Report · Decision Letter 1]

22 Jan 2025

FAM50A acts as a novel prognostic marker and modulates the proliferation of colorectal cancer cells via CylinA2/CDK2 pathway

PONE-D-24-50052R1

Dear Dr. Li,

We’re pleased to inform you that your manuscript has been judged scientifically suitable for publication and will be formally accepted for publication once it meets all outstanding technical requirements.

Kind regards,

Peng Zhang, Ph.D.

Academic Editor

PLOS ONE
---

## [Editor Report · Acceptance letter]

PONE-D-24-50052R1

PLOS ONE

Dear Dr. Li,

I'm pleased to inform you that your manuscript has been deemed suitable for publication in PLOS ONE. Congratulations! Your manuscript is now being handed over to our production team.

Kind regards,

on behalf of

Prof. Peng Zhang

Academic Editor

PLOS ONE